# The role of social support in mitigating the effects of increased screen time on adolescent mental health

William Youkang Zhou[1]*, Luisa Franzini[2]

**1** Walt Whitman High School, Bethesda, Maryland, United States of America, **2** Department of Health Policy and Management, School of Public Health, University of Maryland at College Park, College Park, Maryland, United States of America

* billykzhou@gmail.com

**Data Availability Statement:** The NHIS dataset is publicly available and is available at: https://www.cdc.gov/nchs/nhis/2022nhis.htm Scholars can access and download the dataset from that link in the same manner as the authors.

## Abstract

Excessive screen time has been linked to deteriorating mental health in adolescents, a relationship potentially intensified during the COVID-19 pandemic. Conversely, supportive social environments are associated with improved mental well-being. This study examined the association between screen time, social/emotional support, and mental health among adolescents during the pandemic using data from the 2022 National Health Interview Survey's Sample Child Interview. The analysis focused on adolescents aged 12 to 17. Mental health outcomes included (1) the frequency of feeling sad or depressed and (2) a life satisfaction score. Key independent variables were daily screen time exceeding two hours and consistent receipt of needed social or emotional support. Ordinary least squares regressions with survey weights were applied to ensure nationally representative findings. The study included 2,649 adolescents, with 81% reporting over two hours of daily screen time and 76% consistently receiving social support. Regression results indicated that adolescents with less than two hours of daily screen time and consistent social support reported the lowest frequency of depressive symptoms (coef = 0.79, p<0.001) and the highest life satisfaction scores (coef = 1.34, p<0.001) compared to their peers with greater screen time and/or inconsistent social support. These findings highlight the importance of managing screen time and ensuring robust social support to promote adolescent mental health.

## Introduction

The association between high screen time and overall worse mental health has been well documented [1, 2]. Both high screen time and poor mental health became more prevalent during the COVID-19 pandemic [3]. Studies showed that the average daily screen time for adolescents went up by approximately 52% [4]. A recent study showed a negative association between screen time and the mental health of children 4 to 12 years old during the pandemic [5]. Additionally, there is a strong association between access to social support and improved mental health in adolescents [6, 7]. Social interactions were significantly limited during quarantine, and individuals lacking access to social support experienced worse mental health outcomes

**Funding:** The authors received no specific funding for this work.

**Competing interests:** The authors have declared that no competing interests exist.

[8]. There is a lack of research on how increased screen time during the pandemic impacted adolescent mental health and whether social support can alleviate the impact of screen time on adolescents' mental health.

This study aims to examine the association between screen time, social support, and adolescent mental health during the COVID-19 pandemic. We hypothesize that increased screen time is associated with a higher frequency of feeling depressed, and that consistent receipt of social and emotional support may mitigate this relationship. Adolescence is a critical period of rapid brain development that can have lasting effects on physical or mental health in adulthood. Research has indicated that, during the pandemic, adolescents experienced substantially poor mental health and showed signs of accelerated brain aging [9]. By identifying the variations in adolescent mental health by screen time and social support, the study seeks to understand the importance of regulating screen time and promoting healthy social habits and opportunities for adolescents.

## Methods

### Ethics statement

In accordance with the guidelines provided by the Office for Human Research Protections, this study utilized a secondary public dataset and was therefore exempt from Institutional Review Board (IRB) review. Specifically, the study used the National Health Interview Survey (NHIS) administered by the National Center for Health Statistics (NCHS). The IRB approval and documented consent was obtained from participants by the NCHS Research Ethics Review Board.

### Data

Our study used the 2022 NHIS Sample Child Interview, focusing on adolescents aged 12 to 17. The NHIS annually collects health information from a nationally representative sample of U.S. households, with the child survey addressing topics specifically related to child health. Interviews for the 2022 survey were conducted between January 1, 2022, to December 31, 2022, with responses reflecting this same time period [10]. NHIS has been widely adopted to examine health outcomes and healthcare utilizations [11–13].

### Outcome measures

Our study has two outcome measures: frequency of feeling depressed and life satisfaction score. The frequency of feeling depressed was assessed through parents' responses to the NHIS survey question: "How often does [the adolescent] seem very sad or depressed? Would you say: daily, weekly, monthly, a few times a year, or never?" Responses were recorded on a 5-point Likert scale, with 1 indicating "daily" and 5 indicating "never".

Life satisfaction was assessed based on parents' responses to the NHIS survey question: "Using a scale of 0 to 10, where 0 means "very dissatisfied" and 10 means "very satisfied", how do you think [the adolescent] feels about [the adolescent] life as a whole these days?" Responses were measured on a scale of 1 to 10, with 10 being very happy and satisfied with one's life and 1 being very dissatisfied.

### Key independent variables

The two key independent variables of the study were screen time and access to social support. Screen time was a binary measure, which equaled one if an adolescent had spent over two hours per day on a device for social or entertainment purposes (playing games, watching

programs, using social media, or accessing the internet) and zero otherwise. This measure did not include screen time spent on schoolwork.

Social support was also a binary measure derived from the survey question: "How often does [the adolescent] get the social and emotional support [the adolescent] needs? Would you say always, usually, sometimes, rarely, or never?" We created an indicator that equaled one if an adolescent reported always receiving social support when they needed it and zero otherwise.

### Other independent variables

The other covariates variables included age, sex, race/ethnicity, self-reported general health, family income, parent's highest education, and urban/rural residence. These independent variables have been widely adopted in the literature to examine children's healthcare and health outcomes [14–16].

### Analysis

We first presented adolescents' mental health by screen time and social support and then compared sample characteristics by screen time. Ordinary least squares (OLS) regression with survey weights was applied to estimate the linear association between screen time, social support, and adolescent mental health. OLS regression was appropriate for this study as it captures the continuous and granular nature of these relationships. Survey weights were used to account for the complex sampling methods of the NHIS, so the results of this study were nationally representative [17]. We used the Python packages Pandas and Stats models to run a linear regression and apply survey weights, referencing the NHIS built-in survey weight. We tested different sensitivity analyses, including testing various model specifications and measures, to confirm the results were robust.

Although OLS regressions can capture the granular nature of the relationship, we conducted a sensitivity analysis of categorized variables to address the skewness in the distribution of the outcomes. For the frequency of feeling depressed, we created a binary variable coded as 1 if the response was 'never' and 0 otherwise. For life satisfaction, a binary variable was coded as 1 if the response was 9 or 10 (representing the top 50th percentile of the score range) and 0 otherwise. Logistic regressions were applied. Logistic regressions were applied, and the results were consistent with the main findings and are included in the S1 Table.

The NHIS includes a second measure on availability of community support. The survey question is "Other than parents or adults living in [the adolescent]' home, is there at least one adult in [the adolescent]'s school, neighborhood, or community who makes a positive and meaningful difference in [the adolescent]' life? To our understanding, this question measured availability of support within the community. As a sensitivity analysis, we examined the association between mental health outcomes and this measure of support availability. The results of this analysis are included in the S2 Table.

## Results

The NHIS 2022 Sample Child file had 2,758 participants aged 12 to 17, and 2,649 adolescents have full records, with no missing values of all variables of interest. Approximately 81% of adolescents reported over two hours of screen time and 76% always received social support when they needed it. Fig 1 shows that adolescents who had less than two hours of screen time and always received social support had the lowest frequencies of feeling depressed and the highest life satisfaction scores with a mean of 9.17. Adolescents with over two hours of screen time who didn't always receive social support had the highest frequencies of feeling depressed and the lowest life satisfaction scores with a mean of 7.86.

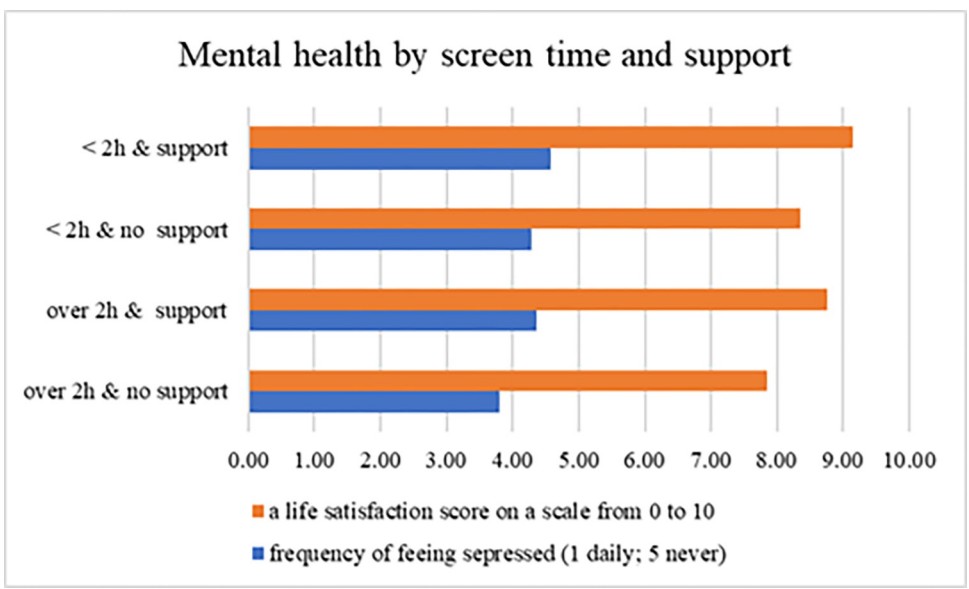

**Fig 1. Mental health by screen time and social support.** Data source: National Health Interview Survey's 2022 Sample Child Interview. The final sample included 2,649 adolescents aged 12 to 17.

Sample characteristics (Table 1) showed that adolescents with over two hours of screen time reported a significantly higher frequency of feeling depressed compared to those with less than two hours (p<0.001). Similarly, life satisfaction scores were significantly lower among adolescents with more than 2 hours of screen time (8.53 vs 9.04, p<0.001). Adolescents with over two hours of screen time were also less likely to reside in rural areas (p<0.001), and their parents were more likely to have fewer years of schooling (p<0.05). However, differences in other characteristics such as race and ethnicity, self-reported health, and income, were not statistically significant.

After controlling for all confounding variables, the regression results (Table 2) indicated that adolescents with over two hours of screen time reported significantly worse mental health compared to those with less than two hours. Specifically, those with over 2 hours of screen time had a higher frequency of feeling depressed (coef = -0.31, p<0.001) and lower life satisfaction scores (coef = -0.54, p<0.001).

The regression results (Table 3) further revealed that adolescents with over two hours of screen time who did not always receive social support reported the worst mental health outcomes. All other groups reported better mental health. Adolescents with less than two hours of screen time and consistent social support reported the least frequency of feeling depressed (coef = 0.79, p<0.001), meaning 0.79 points lower on the frequency of depression scale, and highest life satisfaction scores (coef = 1.34, p<0.001), meaning 1.34 points higher in satisfaction. This group was followed by adolescents with over two hours of screen time who always received social support, and those with less than two hours of screen time but did not always receive social support.

## Discussion

The results of our study showed that screen time was significantly associated with higher frequencies of feeling depressed and lower life satisfaction scores in adolescents. In addition, consistently receiving social support was significantly associated with lower frequencies of feeling

**Table 1. Sample characteristics.**

| | Screen time> = 2hr | | Screen time <2hr | | |
| --- | --- | --- | --- | --- | --- |
| | n = 2,147 | | n = 502 | | |
| | mean | Std. | mean | Std. | p-value |
| Frequency of feeling depressed (1 daily; 5 never) | 4.23 | 1.05 | 4.53 | 0.84 | <0.001 |
| A life satisfaction score on a scale from 0 to 10 | 8.53 | 1.65 | 9.04 | 1.32 | <0.001 |
| Female | 0.49 | 0.50 | 0.50 | 0.50 | 0.90 |
| Age | 14.75 | 1.64 | 14.36 | 1.77 | <0.001 |
| Race/Ethnicity | | | | | |
| Non Hispanic White | 0.50 | 0.50 | 0.50 | 0.50 | 0.84 |
| Non Hispanic Black | 0.11 | 0.31 | 0.09 | 0.28 | 0.21 |
| Hispanic | 0.26 | 0.44 | 0.27 | 0.44 | 0.73 |
| Non Hispanic Asian | 0.07 | 0.25 | 0.08 | 0.27 | 0.40 |
| Other Race/Ethnicity | 0.06 | 0.24 | 0.07 | 0.25 | 0.63 |
| Self-reported Health | | | | | |
| Excellent | 0.83 | 0.38 | 0.86 | 0.35 | 0.10 |
| Great | 0.14 | 0.34 | 0.11 | 0.31 | 0.10 |
| Fair/poor | 0.03 | 0.18 | 0.03 | 0.18 | 0.73 |
| Family Income | | | | | |
| <100% Federal Poverty Line (FPL) | 0.11 | 0.31 | 0.11 | 0.32 | 0.59 |
| 100–200% FPL | 0.20 | 0.40 | 0.18 | 0.38 | 0.17 |
| 200–400% FPL | 0.28 | 0.45 | 0.28 | 0.45 | 0.99 |
| > 400% FPL | 0.41 | 0.49 | 0.43 | 0.50 | 0.45 |
| Parent highest level of education | | | | | |
| No high school | 0.05 | 0.21 | 0.07 | 0.25 | 0.06 |
| High school | 0.31 | 0.46 | 0.26 | 0.44 | 0.04 |
| College degree | 0.41 | 0.49 | 0.39 | 0.49 | 0.62 |
| Graduate School | 0.24 | 0.43 | 0.28 | 0.45 | 0.07 |
| Rural | 0.13 | 0.33 | 0.18 | 0.38 | 0.00 |

Note. Data source: National Health Interview Survey's 2022 Sample Child Interview.

depressed and higher life satisfaction scores. Specifically, adolescents who reported less than 2 hours of screen time and consistently received social support experienced a 0.79-point reduction in depression frequency and a 1.34-point increase in life satisfaction.

Excessive screen time during adolescence has been linked to a variety of negative health outcomes. Studies have found that high screen time is associated with psychosocial problems in adolescence, such as increased depressive symptoms and anxiety, as well as a higher likelihood of being overweight [18]. Moreover, excessive screen time has been shown to disrupt sleep patterns, contributing to shorter sleep duration, delayed sleep onset, and more frequent sleep disturbances [19, 20]. Screen time is also linked to poor attachment to parents and peers, which may delay social and emotional development [21]. Research further indicates that excessive screen use can negatively impact academic performance due to reduced focus and increased procrastination [22]. Additionally, prolonged screen exposure has been associated with reduced physical activity, leading to a sedentary lifestyle and increased risk of metabolic disorders [23]. These findings collectively demonstrate that excessive screen time poses significant risks to well-being, particularly during the critical developmental stage of adolescence [24].

**Table 2. Association between screen time and mental health of adolescents (n = 2,649).**

| | Frequency of feeling depressed (1 daily; 5 never) | | A life satisfaction score on a scale from 0 to 10 | |
|---|---|---|---|---|
| | Coefficient | p | Coefficient | p |
| Screen time below 2hr | ref | | ref | |
| Screen time over 2hr | -0.31 | <0.001 | -0.54 | <0.001 |
| Female | -0.37 | <0.001 | -0.20 | 0.01 |
| Age | -0.04 | <0.001 | -0.05 | 0.02 |
| Race/Ethnicity | | | | |
| Non Hispanic White | ref | | ref | |
| Non Hispanic Black | 0.33 | <0.001 | 0.44 | <0.001 |
| Hispanic | 0.32 | <0.001 | 0.75 | <0.001 |
| Non Hispanic Asian | 0.22 | <0.001 | 0.52 | <0.001 |
| Other Race/Ethnicity | -0.13 | 0.25 | -0.13 | 0.34 |
| Family Income | | | | |
| <100% FPL | ref | | ref | |
| 100–200% FPL | 0.17 | 0.07 | 0.00 | 1.00 |
| 200–400% FPL | 0.14 | 0.16 | -0.04 | 0.77 |
| > 400% FPL | 0.15 | 0.13 | 0.03 | 0.86 |
| Parent highest level of education | | | | |
| No high school | ref | | ref | |
| High school | -0.13 | 0.24 | 0.05 | 0.79 |
| College degree | -0.15 | 0.21 | -0.05 | 0.81 |
| Graduate School | -0.08 | 0.54 | 0.05 | 0.79 |
| Self-reported Health | | | | |
| Fair/poor | ref | | ref | |
| Excellent | 1.10 | <0.001 | 2.07 | <0.001 |
| Great | 0.72 | <0.001 | 1.35 | <0.001 |
| Rural | 0.02 | 0.79 | 0.05 | 0.70 |
| Constant | 4.15 | <0.001 | 7.65 | <0.001 |

Note. Data source: National Health Interview Survey's 2022 Sample Child Interview. Sampling weights were applied. Results were nationally representative.

The American Academy of Pediatrics (AAP) recommends parents to limit children and adolescents' screen time to two hours per day [25]. These guidelines provide a reference for parents to monitor screen time. Our study findings suggest that in addition to monitoring and regulating screen time, it is equally important that adolescents maintain a healthy social life

**Table 3. Association between screen time and with social support and mental health of adolescents (n = 2,649).**

| | Frequency of feeling depressed (1 daily; 5 never) | | Life satisfaction score on a (scale from 0 to 10) | |
|---|---|---|---|---|
| | Coefficient | p | Coefficient | p |
| Screen time and social support | | | | |
| over 2h & no support | reference | | reference | |
| < 2h & support | 0.79 | <0.001 | 1.34 | <0.001 |
| over 2h & support | 0.56 | <0.001 | 0.84 | <0.001 |
| < 2h & no support | 0.43 | <0.001 | 0.34 | 0.11 |

Note. Data source: National Health Interview Survey's 2022 Sample Child Interview. All the covariates were controlled for. Sampling weights were applied. Results were nationally representative.

and stay in contact with friends. Given that adolescents spend the majority of their time at school or at home, schools, communities, and parents should collaborate to develop strategies and organize activities or social events that prioritize adolescent mental health [26].

The study has several limitations. First, mental health measures were reported by parents instead of adolescents themselves, which might not accurately capture adolescents' own perceptions. Adolescents often conceal mental health issues due to the stigma surrounding teen mental health [27]. Second, the survey used a cut-off of 2 hours of daily screen time for social or entertainment purposes, which limits the ability to study extensive screen time users. Third, social and emotional support was measured solely by the frequency with which adolescents received it. Future surveys could collect more detailed information on the types and sources of support provided. Finally, we used the cross-sectional analysis, which does not allow for causal inference. Hence the results can't interpret with the causal impact.

Despite these limitations, our study has several strengths. It is a nationally representative study that examines the association between screen time and mental health among adolescents. Moreover, it highlights the role of social and emotional support as a modifiable factor in promoting positive mental health, which could inform future actionable interventions.

## Conclusion

Overall, our results indicate that high screen time is associated with higher frequencies of feeling depressed and lower life satisfaction scores among adolescents. On the other hand, receiving social support was associated with lower frequencies of feeling depressed and higher life satisfaction scores. Communities, schools, parents, and adolescents should work together to discourage excessive screen time and encourage social interaction, creating an environment that promotes adolescent mental health.

## Supporting information

**S1 Table. Logistic regressions.**
(DOCX)

**S2 Table. Availability of community support.**
(DOCX)

## Author Contributions

**Conceptualization:** Luisa Franzini.

**Data curation:** William Youkang Zhou.

**Formal analysis:** William Youkang Zhou.

**Methodology:** William Youkang Zhou, Luisa Franzini.

**Project administration:** Luisa Franzini.

**Supervision:** Luisa Franzini.

**Validation:** Luisa Franzini.

**Writing – original draft:** William Youkang Zhou.

**Writing – review & editing:** William Youkang Zhou, Luisa Franzini.

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
