## [Decision Letter · Decision Letter 0]

16 Sep 2024

PMEN-D-24-00336

The Role of Social Support in Mitigating the Effects of Increased Screen Time on Adolescent Mental Health

PLOS Mental Health

Dear Dr. Zhou,

Thank you for submitting your manuscript to PLOS Mental Health. After careful consideration, we feel that it has merit but does not fully meet PLOS Mental Health’s publication criteria as it currently stands. Therefore, we invite you to submit a revised version of the manuscript that addresses the points raised during the review process.

EDITOR: I have completed my evaluation of your manuscript. The reviewers recommend reconsideration of your manuscript following revision. I invite you to resubmit your manuscript after addressing the comments below.

We look forward to receiving your revised manuscript.

Kind regards,

Gellan Karamallah Ramadan Ahmed

Academic Editor

PLOS Mental Health

Journal Requirements:

1. We have amended your Competing Interest statement to comply with journal style. We kindly ask that you double check the statement and let us know if anything is incorrect. 

2. Please provide a/amend your detailed Financial Disclosure statement. This is published with the article. It must therefore be completed in full sentences and contain the exact wording you wish to be published.

**Please only choose the relevant sentences from below**

1. Please clarify all sources of funding (financial or material support) for your study. List the grants (with grant number) or organizations (with url) that supported your study, including funding received from your institution. 

2. State the initials, alongside each funding source, of each author to receive each grant.

3. State what role the funders took in the study. If the funders had no role in your study, please state: “The funders had no role in study design, data collection and analysis, decision to publish, or preparation of the manuscript.”

4. If any authors received a salary from any of your funders, please state which authors and which funders.

If you did not receive any funding for this study, please simply state: “The authors received no specific funding for this work."

3. Please provide separate figure files in .tif or .eps format.

https://journals.plos.org/mentalhealth/s/figures 

https://journals.plos.org/mentalhealth/s/figures#loc-file-requirements 

4. We note that your Data Availability Statement is currently as follows: "Dataset is publicly available"

Reviewers' comments:

Reviewer's Responses to Questions

**Comments to the Author**

1. Does this manuscript meet PLOS Mental Health’s publication criteria? Is the manuscript technically sound, and do the data support the conclusions? The manuscript must describe methodologically and ethically rigorous research with conclusions that are appropriately drawn based on the data presented.

Reviewer #1: Partly

Reviewer #2: Partly

2. Has the statistical analysis been performed appropriately and rigorously?

Reviewer #1: No

Reviewer #2: No

3. Have the authors made all data underlying the findings in their manuscript fully available (please refer to the Data Availability Statement at the start of the manuscript PDF file)?

Reviewer #1: No

Reviewer #2: Yes

4. Is the manuscript presented in an intelligible fashion and written in standard English?

Reviewer #1: No

Reviewer #2: Yes

5. Review Comments to the Author

Reviewer #1: Regarding the "background", improve the definition and delimitation of the variables: The term "social/emotional support" should be better defined. What does it mean in this context? What type of social support is being considered (family, peer, institutional or other)?

Specifying the temporal context: When you mention "during/after the COVID-19 pandemic", it is necessary to clearly delimit the phases you are referring to. Define whether we are talking about confinement, reopening, or both.

Objectives and approach of the study: The objective of the study is mentioned in a general way. It would be useful to specify what we are looking to measure, how screen time is expected to affect mental health, and how social support can moderate that relationship. Clarity and precision in the presentation of the method: Although the variables and techniques used are mentioned, important details are missing that increase the clarity and replicability of the study. For example, more information needs to be provided about sample selection, how data were recognized, and whether potentially confounding variables are controlled. Furthermore, it is not detailed how "social or emotional support" was measured.

Further development in outcome measures: The description of the dependent variables is vague. You should explain what the "life satisfaction" measure is and how the "frequency of children who seem sad or depressed" is calculated. It is important that measurement tools are validated and accurately referenced to ensure scientific quality.

Justification of the statistical analysis: Although it mentions ordinary least squares (OLS) regression, it does not explain why this technique was selected and whether possible violated assumptions (such as heteroscedasticity or multicollinearity) are considered. Additionally, it would be helpful to explain how the survey weights were performed and why they were necessary for this study.

Details of the population context: When indicating that the results are "nationally representative", it would be relevant to specify which country or region was taken into account and how this representativeness is ensured. National representativeness often requires a specific sampling design, which should be detailed.

that's all

Reviewer #2: I’ve read the work titled “The Role of Social Support in Mitigating the Effects of Increased Screen Time on Adolescent Mental Health”.

The introduction is well written and summarizes the current state of the art well. The literature used is appropriate and recent.

The methods are generally well written. Nevertheless, I would add more details to some passages in order to make this part more informative. I recommend adding more information regarding the population from which the study sample was drawn and adding more details regarding the cohort selection. It might be helpful to add a flow chart illustrating the cohort selection procedure. Having this information would make it easier for the study to be replicated. In any case, I think the methods section is a bit lacking. I recommend reviewing this part better.

In the Results section, where appropriate, in the various titles of the tables I recommend adding more information such as, for example, the time period to which the data refers.

The discussion is generally well written. Particularly, I appreciated the section on limitations of the study. Nonetheless, I recommend discussing the weaknesses in more detail (e.g. adding any bias related to the data source). Finally, I recommend including a series of strengths related to the data source or study design that was used in order to give greater value to the work.

6. PLOS authors have the option to publish the peer review history of their article (what does this mean?). If published, this will include your full peer review and any attached files.

**Do you want your identity to be public for this peer review?** For information about this choice, including consent withdrawal, please see our Privacy Policy.

Reviewer #1: No

Reviewer #2: No

---

## [Decision Letter · Decision Letter 1]

12 Nov 2024

PMEN-D-24-00336R1

The Role of Social Support in Mitigating the Effects of Increased Screen Time on Adolescent Mental Health

PLOS Mental Health

Dear Dr. Zhou,

Thank you for submitting your manuscript to PLOS Mental Health. After careful consideration, we feel that it has merit but does not fully meet PLOS Mental Health’s publication criteria as it currently stands. Therefore, we invite you to submit a revised version of the manuscript that addresses the points raised during the review process.

We look forward to receiving your revised manuscript.

Kind regards,

Gellan Karamallah Ramadan Ahmed

Academic Editor

PLOS Mental Health

Journal Requirements:

Additional Editor Comments (if provided):

I have completed my evaluation of your manuscript. The reviewers recommend reconsideration of your manuscript following revision. I invite you to resubmit your manuscript after addressing the comments below.

Reviewers' comments:

Reviewer's Responses to Questions

**Comments to the Author**

1. If the authors have adequately addressed your comments raised in a previous round of review and you feel that this manuscript is now acceptable for publication, you may indicate that here to bypass the “Comments to the Author” section, enter your conflict of interest statement in the “Confidential to Editor” section, and submit your "Accept" recommendation.

Reviewer #1: All comments have been addressed

Reviewer #2: All comments have been addressed

2. Does this manuscript meet PLOS Mental Health’s publication criteria? Is the manuscript technically sound, and do the data support the conclusions? The manuscript must describe methodologically and ethically rigorous research with conclusions that are appropriately drawn based on the data presented.

Reviewer #1: Yes

Reviewer #2: Yes

3. Has the statistical analysis been performed appropriately and rigorously?

Reviewer #1: Yes

Reviewer #2: Yes

4. Have the authors made all data underlying the findings in their manuscript fully available (please refer to the Data Availability Statement at the start of the manuscript PDF file)?

Reviewer #1: Yes

Reviewer #2: Yes

5. Is the manuscript presented in an intelligible fashion and written in standard English?

Reviewer #1: Yes

Reviewer #2: Yes

6. Review Comments to the Author

Reviewer #1: Specify the statistical aspect of the research work, if the study had the objective of estimating the association between Screen time, social/emotional support, and mental health among adolescents. what is the level

during/after the COVID-19 pandemic. The Conclusion is clear, the existence of an association between the study variables was determined. BUT IT WOULD BE IMPORTANT TO SPECIFY THE RESULTS

(the level, degree or strength of the association).

Reviewer #2: I have read the article titled “The Role of Social Support in Mitigating the Effects of Increased Screen Time on Adolescent Mental Health” and am pleased that the authors have addressed the issues raised in the previous round.

Currently, the manuscript is a well-written research paper whose objective is to identify profiles of prompt, adequate and continuous physician follow-up care and other physician practice features, and to estimate the association between screen time, social/emotional support, and mental health among adolescents during/post the COVID-19 pandemic.

I believe the manuscript meets the publication standards of the journal.

7. PLOS authors have the option to publish the peer review history of their article (what does this mean?). If published, this will include your full peer review and any attached files.

**Do you want your identity to be public for this peer review?** For information about this choice, including consent withdrawal, please see our Privacy Policy.

Reviewer #1: No

Reviewer #2: No

---

## [Decision Letter · Decision Letter 2]

27 Nov 2024

PMEN-D-24-00336R2

The Role of Social Support in Mitigating the Effects of Increased Screen Time on Adolescent Mental Health

PLOS Mental Health

Dear Dr. Zhou,

Thank you for submitting your manuscript to PLOS Mental Health. After careful consideration, we feel that it has merit but does not fully meet PLOS Mental Health’s publication criteria as it currently stands. Therefore, we invite you to submit a revised version of the manuscript that addresses the points raised during the review process.

We look forward to receiving your revised manuscript.

Kind regards,

Gellan Karamallah Ramadan Ahmed

Academic Editor

PLOS Mental Health

Journal Requirements:

Additional Editor Comments (if provided):

Reviewers' comments:

Reviewer's Responses to Questions

**Comments to the Author**

1. If the authors have adequately addressed your comments raised in a previous round of review and you feel that this manuscript is now acceptable for publication, you may indicate that here to bypass the “Comments to the Author” section, enter your conflict of interest statement in the “Confidential to Editor” section, and submit your "Accept" recommendation.

Reviewer #1: All comments have been addressed

2. Does this manuscript meet PLOS Mental Health’s publication criteria? Is the manuscript technically sound, and do the data support the conclusions? The manuscript must describe methodologically and ethically rigorous research with conclusions that are appropriately drawn based on the data presented.

Reviewer #1: Yes

3. Has the statistical analysis been performed appropriately and rigorously?

Reviewer #1: Yes

4. Have the authors made all data underlying the findings in their manuscript fully available (please refer to the Data Availability Statement at the start of the manuscript PDF file)?

Reviewer #1: Yes

5. Is the manuscript presented in an intelligible fashion and written in standard English?

Reviewer #1: Yes

6. Review Comments to the Author

Reviewer #1: Observations raised.

7. PLOS authors have the option to publish the peer review history of their article (what does this mean?). If published, this will include your full peer review and any attached files.

**Do you want your identity to be public for this peer review?** For information about this choice, including consent withdrawal, please see our Privacy Policy.

Reviewer #1: No

---

## [Editor Report · Decision Letter 3]

4 Dec 2024

The Role of Social Support in Mitigating the Effects of Increased Screen Time on Adolescent Mental Health

PMEN-D-24-00336R3

Dear Mr. Zhou,

We are pleased to inform you that your manuscript 'The Role of Social Support in Mitigating the Effects of Increased Screen Time on Adolescent Mental Health' has been provisionally accepted for publication in PLOS Mental Health.

Best regards,

Gellan Karamallah Ramadan Ahmed

Academic Editor

PLOS Mental Health